# Why Do We Need JAK Inhibitors in Systemic Lupus Erythematosus?

**DOI:** 10.3390/ijms231911788

**Published:** 2022-10-04

**Authors:** Patricia Richter, Anca Cardoneanu, Alexandra Maria Burlui, Luana Andreea Macovei, Ioana Bratoiu, Oana Nicoleta Buliga-Finis, Elena Rezus

**Affiliations:** 1Department of Rheumatology, “Grigore T Popa” University of Medicine and Pharmacy, 700115 Iasi, Romania; 2Clinical Rehabilitation Hospital, 700661 Iasi, Romania; 3Department of Internal Medicine, “Grigore T Popa” University of Medicine and Pharmacy, 700115 Iasi, Romania; 4“Sf. Spiridon” Clinical Emergency Hospital, 700111 Iasi, Romania

**Keywords:** systemic lupus erythematosus, Janus kinases, JAK inhibitor, cytokines

## Abstract

Systemic lupus erythematosus (SLE) is a chronic, multifactorial autoimmune disease with complex pathogenesis characterized by the imbalance of pro-inflammatory and anti-inflammatory cytokines. Janus kinases (JAKs), intracellular non-receptor tyrosine kinases, are essential for signal pathways of many cytokines. The JAK signal transducers and activators of transcription (STAT) pathways consist of four JAK kinases and seven STATs family members. The dysregulation of JAK-STAT pathways represents an important process in the pathogenesis of SLE. Thus, the use of therapies that target specific signaling pathways would be a challenge in SLE. It is well known that JAK inhibitors have real potential for the treatment of rheumatic diseases, but their efficacy in the treatment of SLE remains to be determined. JAK inhibitors are currently being investigated in phase II and III trials and are considered to become the next stage in SLE therapy. In this review, we report the current data regarding the efficacy of JAK inhibitors in SLE. The development of clinically useful kinase inhibitors might improve upon traditional therapeutic strategies.

## 1. Introduction

Systemic lupus erythematosus (SLE) is a chronic systemic autoimmune disease characterized by complex pathogenesis that includes genetic factors, environmental triggers, and hormone molecules, as well as an overproduction of an array of cytokines [1]. Under these conditions a loss of self-tolerance and an overproduction of autoantibodies took place. Both innate and adaptive immune systems have important roles in SLE pathogenesis [1].

Type 1 interferons (IFN α and ß, mostly released by dendritic cells) and type 2 IFN (IFN γ -mostly secreted by T cells) are present in SLE. Moreover, an elevated expression of type 1 IFN-regulated genes, known as IFN signature, is considered to be the main characteristic of SLE [1]. The relationship between JAK/STAT (Janus kinase and Signal Transducers and Activators of Transcription) and IFNs pathway is based on the discovery of type 1 and type 2 IFNs mechanism of action [1,2]. They use JAK/STAT cascade signaling pathway to exert their biological actions [1]. Analysis of the inhibition of the JAK/STAT pathway has shown that it has a central role in the decrease of SLE inflammation [1].

Existing treatments such as glucocorticoids and immunosuppressive drugs can be associated with various severe side effects and incomplete efficacy [3,4]. Under these circumstances, SLE remains a condition with high morbidity and mortality [3,5]. Biological molecules targeting proinflammatory cytokines have changed the treatment of autoimmune diseases [6]. Thus, an upcoming target for SLE is the JAK/STAT pathway [7]. Many JAK inhibitors have been studied for the treatment of SLE [8]. Regarding the recent information concerning cytokine signaling, the studies analyze if this intracellular signaling can have a safe and efficient role in SLE patients [6]. Therefore, an update on these drugs’ development is essential [9,10].

Cytokines and cell surface molecules which act through the JAK/STAT pathway play a pivotal role in the pathogenesis of inflammatory autoimmune diseases [11,12]. These can transduce a diversity of intracellular signals by binding to receptors, generating cell functions, and the transcription of new cytokines [8]. The evidence of the promising efficacy of the JAK/STAT pathway in type I and II cytokine signaling has increased the research in the field of rheumatic diseases [12,13]. Thus, their targeted inhibition can be associated with disease control [14]. 

The first JAK inhibitor approved for human use was tofacitinib, a JAK1/3 inhibitor [1]. Baricitinib, which selectively inhibits JAK1 and JAK2, is also under investigation [3,15]. In December 2018, Food and Drug Administration (FDA) granted the “Fast Track” status for baricitinib use in SLE treatment [10]. Next-generation JAK inhibitor agents such as upadacitinib and filgotinib are still being evaluated in clinical trials [16].

The purpose of this paper is to assess the current evidence from case reports and clinical trials regarding JAK inhibitor efficacy in SLE treatment [9].

## 2. JAK–STAT Pathway

The JAK/STAT pathways are represented by four JAK kinases (JAK1, JAK2, JAK3, and non-receptor tyrosine-protein kinase 2-TYK2) and seven STAT family members (STAT1, STAT2, STAT3, STAT4, STAT 5a, STAT 5b, STAT6) [11,14,17,18]. Their different combinations induce the transcription of various genes via STATs [8]. The enzymes whose role is to phosphorylate signaling molecules are called kinases. Of 518 kinases, JAK is classified as a typical tyrosine kinase [8] that transfer phosphates from adenosine triphosphate (ATP) to tyrosine residues on other proteins such as cytokine receptors or even JAKs [12,19]. Protein kinases are essential regulators of cellular functions. Intracellular signal transduction is realized through the connection between JAK, TYK2 isoforms, and STAT members [8]. The activity of every JAK depends on selective interactions with cytokine receptors. Moreover, each cytokine receptor provides a specific combination with JAK kinases, which has a crucial implication in therapeutic maneuvers [11,20].

JAKs receive signals from various cytokine receptors of interleukin (IL) and IFN members [13,21]. Particularly, the role of JAKs in signaling is focused on a subset of cytokines that use type I and II cytokine receptors. There are described more than 50 soluble molecules, including IL-2, IL-3, IL-4, IL-5, IL-6, IL-12, IFNs, endocrine factors (including growth hormone, prolactin and leptin). colony-stimulating factors (erythropoietin, thrombopoietin and granulocyte–macrophage colony-stimulating factor (GM-CSF)). Their effects are produced after the combination with a specific JAK [14]. TYK2 is activated after the receptor is bound by IL-23, IL-12 and type I IFNs. Specific combinations of STAT family members with their receptors will be phosphorylated by JAKs, leading to STAT dimerization, nuclear translocation and gene regulation [21].

### 2.1. Mechanism of JAK-STAT Signaling Pathway

Several extracellular cytokines and growth factors bind to their specific receptors on targeted cells and initiate a cascade of intracellular signals. JAKs attach to the cytoplasmic domains of these cytokine receptors. This results in subsequent rearrangement of the receptor subunits which is called “cytokine receptor engagement”. This mechanism leads to intracellular trans-phosphorylation of receptor-associated JAKs [11,12]. The phosphorylation process enables JAKs activation. Consecutively, they phosphorylate each other, as well as the intracellular components of the receptors, leading to the emergence of docking sites [11,12,14]. This enables the recruitment of transcription factors of the STAT family, allowing them to bind to the receptor and become phosphorylated by JAKs. Then, the phosphorylated STAT homo- or heterodimers enter the nucleus. Here, STAT induces the transcription of targeted genes [11,14,22]. This heterogenicity of STAT dimerization can produce many biological responses [11].

### 2.2. Inhibition of JAK-STAT Signaling Pathway

JAK inhibitors are the newest class of targeted synthetic disease-modifying antirheumatic drugs (tsDMARDs). These are orally small molecules that are able to enter into cellular cytoplasm and directly modulate intracellular signaling [14]. JAK inhibitors suppress the STAT pathways and inhibit the effects of many proinflammatory cytokines [14]. In particular, tsDMARDs act by blocking the ATP-binding site in the catalyzing region of the JAK enzymes [1,23]. This suppression of downstream signaling pathways is associated with immunomodulatory consequences [14]. In addition, blocking the phosphorylation of cytokine receptors and gene transcription can lead to impaired differentiation of Th1, Th2, and Th17 cells [12,19].

Type I IFN-mediated monogenic autoinflammatory conditions (type I interferonopathies) consist of a heterogeneous group of inflammatory autoimmune diseases characterized by the activation of the type I IFN axis. This multifactorial activation has been also highlighted in SLE. The major signaling pathway activated by IFNs is the JAK/STAT pathway, so its inhibition can be a good therapeutic solution [24,25,26]. Treatments that target JAK/STAT pathways have the potential to show positive effects by minimizing the use of glucocorticoids and immunosuppressant drugs [14]. JAK inhibitors improved clinical symptoms and laboratory parameters, decreased flares, and the expression of IFN-stimulated genes [24]. The use of JAK inhibitors and their potential effectiveness in SLE are currently being investigated.

In the last years, several drugs which inhibit different proteins of the JAK/STAT pathway are under study. Some of them act exclusively on a single site (filgotinib-JAK1) or have effects on multiple sites (tofacitinib-JAK1 and JAK3; ruxolitinib and baricitinib-JAK1 and JAK2). Another classification divided the molecules into first-generation pan-JAK inhibitors (tofacitinib, baricitinib, ruxolitinib, peficitinib) and second-generation selective JAK inhibitors (decernotinib, filgotinib, upadacitinib) [12,24,27,28].

Cytokines that activate the innate and adaptive immune system such as type I IFNs, IL-12, IL-23, and those that activate T–B cell interaction: IL-21, IL-6, and IL-4 are considered to be potential targets of JAK inhibition in SLE [14,16,29]. The disrupted regulation between type I and II IFNs and B cells are two major signatures in SLE, the former being targeted by JAK inhibitors [12]. For instance, the inhibition of JAK1 suppresses both IL-6 and type I IFN signaling [30]. Moreover, cytokines like tumor necrosis factor-alpha (TNF-α), IL-2, IL-10, IL-15, IL-17, and B cell activating factor (BAFF) play prominent roles in SLE pathogenesis. The main intracellular mechanisms of action are represented in Figure 1.

### 2.3. Main Inhibitors of JAK/STAT Pathway

#### 2.3.1. Tofacitinib

Tofacitinib, the first tested JAK inhibitor, has a high selectivity for JAK1 and JAK3. It is less effective for the inhibition of JAK2 and has limited action on TYK2 [6,31,32].

Regarding the genetic risk haplotype, results from a pilot phase Ib/IIa, double-blind trial confirmed that the SLE immunological response to tofacitinib depends on STAT4 risk allele rs7574865[T]. This allele is associated with severe SLE symptoms. Furthermore, tofacitinib showed significantly positive outcomes in cardiometabolic markers such as high-density lipoprotein cholesterol (HDL-C) levels [29,33,34,35,36]. In SLE patients positive for STAT4 risk allele, tofacitinib led to a decreased expression of interferon-response genes and low levels of circulating density granulocytes and neutrophil NETosis, as well as to the suppression of pSTAT1 in CD4+ T cells [36].

Evidence from murine SLE models has demonstrated that tofacitinib modulated lupus-associated parameters levels such as antinuclear antibody (ANA), anti-double-stranded deoxyribonucleic acid (anti-dsDNA) antibodies, proteinuria, skin rash, and type I IFN related responses. Regarding the lipoprotein profile in tofacitinib-treated mice, the authors highlighted the reduction of free cholesterol levels. Tofacitinib is considered to be a “vasculoprotective” agent [12,37,38,39,40,41,42].

Similar results were seen in a rhupus patient (coexistence of SLE and rheumatoid arthritis) with class III glomerulonephritis (GN), and joint and skin involvement. The patient treated with tofacitinib 5mg twice daily and a decrease of cortison dose was also reached, in accordance with 2019 EULAR recommendations [43,44].

The first report showing the efficacy of tofacitinib in 10 SLE patients was performed by You et al. The results were promising for arthritis and cutaneous manifestations but were limited for serological markers [45].

Another study used the Cutaneous Lupus Erythematosus Disease Area and Severity Index (CLASI) score to evaluate skin manifestations. In 3 “recalcitrant” cutaneous lupus patients, the outcomes were encouraging as the CLASI score showed a significant improvement [46]. Additionally, a recent case report confirmed the favorable tofacitinib effects on SLE refractory alopecia. The patient experienced hair regrowth after being treated with a JAK inhibitor [40].

Another case report highlighted a decrease of anti-dsDNA antibody levels to normal values after tofacitinib treatment in a patient with SLE associated with rheumatoid arthritis [47].

Another report presented the cases of two patients with cold-induced finger erythematous lesions. The patients suffered from familial chilblain lupus (FCh-L), which is a monogenic autosomal dominant form of cutaneous lupus erythematosus. This was caused by mutations in the nucleases 3–5′ repair exonuclease 1 (TREX1) or SAMHD1 and is associated with systemic skin and biological involvement due to type I IFN signature. Currently, there is no effective treatment available, but it seems that tofacitinib can induce a strong suppression, leading to lower discomfort and pain in these patients [40,48].

However, tofacitinib’s use in SLE patients still has to be defined and it is now being tested in two phase I/II clinical trials regarding skin manifestation. In the NCT03159936 study, adult individuals with discoid lupus are investigated. The NCT03288324 study is currently recruiting young adults with moderate to severe cutaneous lupus erythematosus (CLE) [49,50,51].

The main ongoing studies on the use of tofacitinib in SLE patients are presented in Table 1 and Table 2. They include the study design, the primary and secondary endpoints, as well as the most important statistically significant results.

#### 2.3.2. Baricitinib

Baricitinib is an orally administered, low-weight molecule, which selectively inhibits JAK1 and JAK2 subtypes. Baricitinib received its first global approval in the European Union (EU) on the 13 February 2017 for the treatment of moderate to severe active rheumatoid arthritis patients who did not show a proper response or were intolerant to DMARDs [15,52].

In a phase IIb clinical trial across 11 countries, Wallace et al. presented the role of baricitinib in patients with active non-renal SLE. These individuals had skin and joint manifestations, two of the most common clinical expressions, and were non-responsive to standard treatments. Significantly more subjects in the baricitinib group (in particular, the 4 mg dose) achieved a resolution of arthritis or skin rash (according to SLEDAI-2K criteria) after 24 weeks [3,14,53]. In addition, the patients on baricitinib treatment achieved a higher response according to the SLE Responder Index (SRI) criteria versus placebo [1,3,14]. There were significant improvements in secondary outcomes, including the physician global assessment (PGA), lupus low disease activity state (LLDAS), risk of flares measured by the SSFI, joint tenderness evaluated by 28-joint examination, Worst Joint Pain NRS, and Worst Pain NRS in the baricitinib arm versus placebo. This was the first randomized controlled trial that demonstrated good efficacy of JAK 1/2 inhibition in SLE and an improvement in the quality of patient life. Unfortunately, there were no notable positive changes seen regarding skin lesions [3,54]. Concerning safety issues, side effects were observed in 65% of patients in the placebo group versus 71% of patients in the 2 mg baricitinib group versus 73% of patients in the 4 mg baricitinib group [3,55].

Analyzing the same group of patients, Dörner et al. showed that baricitinib induces modifications in the RNA expression of genes linked to the JAK/STAT pathway [56].

Focusing on specific organ manifestations, two different studies reported improvement of diffuse non-scarring alopecia and refractory papulosquamous rash after baricitinib treatment [57,58]. The same results regarding the positive effects of baricitinib on severe subacute lesions were described by Joos et al. [59].

JAK/STAT signaling is implicated substantially in inflamed CLE skin. Specifically, keratinocytes and dermal immune cells express the activated form of phospho-JAK1. The JAK1/2 and JAK 1/3 inhibitors exert their role by importantly decreasing the expression of CLE-typical chemokines in vitro [60,61,62].

Moreover, a case report showed promising results with JAK1/2 inhibitor. Subacute cutaneous lupus erythematosus (SCLE) with frontal fibrosing alopecia (FFA) presented a complete remission after 2 months of barcitinib therapy [63].

In particular, all patients with FCh-L (familial chilblain lupus) due to TREX1 deficiency from Zimmermann et al. study who were treated with baricitinib accomplished notable improvement in cutaneous manifestations and relief of joint pain. The most important side effects were repeated mild respiratory infections [24,64].

Data of a rhupus subject successfully treated with a JAK1/2 inhibitor was presented by Garufi et al. Baricitinib 4 mg/day induced complete renal remission and controlled joint manifestations in a rhupus patient with class V GN [43].

In a murine model, baricitinib improved renal inflammation, leading to the recovery of the structural proteins in podocytes. This phenomenon was explained through a direct pathogenic effect of INFα whereby the differentiation and maturation of podocytes is inhibited, which consequently induces podocytes loss [43,65,66].

There are also two trials phase III (BRAVE I-NCT03616912 and BRAVE II NCT03616964), in which the efficacy of baricitinib in SLE is under investigation [67,68]. Furthermore, the long-term safety of baricitinib is being studied in a phase III trial SLE-BRAVE-X (NCT03843125) [69].

Table 3 and Table 4 summarize the main case reports and clinical trials using baricitinib in SLE patients. The main endpoints and the most significant results are also described.

#### 2.3.3. Other New Therapies under Study

##### Ruxolitinib

Data from murine lupus models indicated a significant improvement in skin modifications in MRL/lpr mice treated with the selective JAK1/2 inhibitor ruxolitinib [70].

Wenzel et al. provided further support for the therapeutic implications of ruxolitinib, presenting a successfully controlled Chilblain Lupus Erythematosus (ChLE) [7,71].

Consistent results with those highlighted in patients with ChLE treated with JAK1/2 inhibitors were shown by Briand et al. focusing on skin vasculopathy. Thus, a fast clinical improvement with almost complete resolution of skin lesions was described in a patient with FChL due to TREX1 deficiency associated with Aicardi-Goutières syndrome [7,60,72].

In addition, ruxolitinib has been revealed to be effective in blocking the anti-extractable nuclear antigen (ENA) and anti-dsDNA antibody production in SLE patients [73].

##### Solcitinib

A selective JAK1 inhibitor (GSK2586184) had been proposed as a novel therapeutic agent in a phase IIb study that investigated patients with moderate-to-severe active SLE without renal or cerebral involvement who have failed standard therapy. However, the results were not satisfying, and it was prematurely stopped. Notably, the study on solcitinib pointed out important side effects such as drug reaction with eosinophilia and systemic symptoms (DRESS) and hepatic function modifications, as well as a lack of predicted response [74,75,76,77].

##### R333

R333 (R932333), a topical JAK1/3/SYK (Janus kinase and spleen tyrosine kinase) inhibitor, was studied regarding SLE skin involvement. It also failed to achieve its intended aim of demonstrating positive results in discoid lupus with erythema and scaling possibly due to poor penetration [1,45,78,79].

##### Brepocitinib

A large phase IIb trial of brepocitinib, an inhibitor of JAK1 and TYK2, is ongoing for active SLE [80].

##### Filgotinib

Filgotinib, a highly selective JAK1 inhibitor, is the subject of a phase II clinical trial for active CLE (NCT03134222), being evaluated in comparison with lanraplenib (a Syk kinase inhibitor) [33,81]. Filgotinib is also being evaluated in lupus membranous nephropathy (NCT03285711) [82].

##### Upadacitinib

In addition to the above, upadacitinib’s role in SLE has to be defined. A phase II trial is underway to investigate the safety and efficacy of the JAK1 selective inhibitor in SLE in monotherapy or in combination with a Bruton’s tyrosine kinase (BTK) inhibitor-elsubrutinib (NCT03978520) [83].

##### Deucravacitinib

Deucravacitinib is a highly selective TYK2 inhibitor having minimal or no activity against JAK1-3. Two clinical trials referring to the use of deucravacitinib to treat SLE patients were identified: PAISLEY LN NCT03943147 which evaluated participants with lupus nephritis stopped due to an insufficient number of subjects [84] and PAISLEY SLE NCT03252587 [85] a long-term trial on safety and efficacy NCT03920267 [86].

The main clinical trials, as well as their preliminary results regarding new JAK inhibitors used in SLE treatment, are described in Table 5.

## 3. Conclusions

Despite promising results from clinical trials, none of these therapies have managed to receive approval to be used in clinical practice. Thus far, the best results have been obtained from the use of tofacitinib and baricitinib.

Due to its complicated pathogenesis, the management of SLE is still changeling. While the therapeutic use of JAK inhibitors has been demonstrated, their role in SLE remains to be determined. Despite the safety and efficacy profiles showing encouraging results, future clinical trials are necessary. All these data published so far represent a cornerstone for future studies that we hope will bring new useful information regarding the therapy of SLE patients.

## Figures and Tables

**Figure 1 ijms-23-11788-f001:**
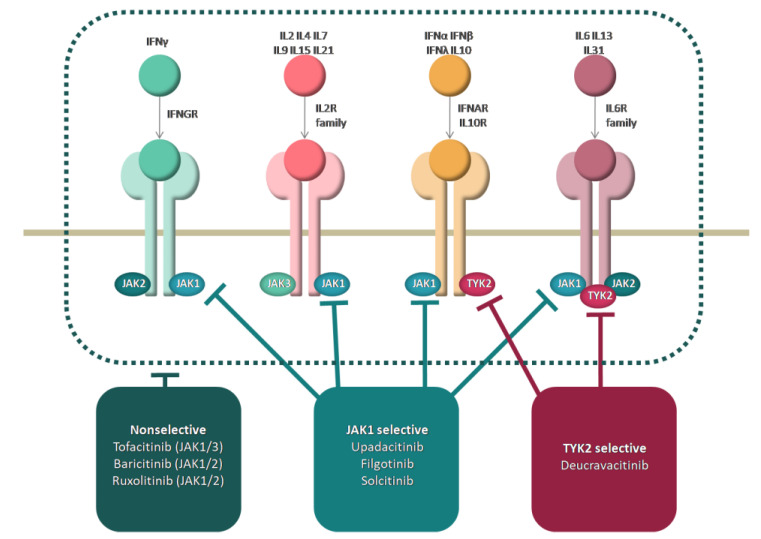
Mechanism of action of JAK inhibitors.

**Table 1 ijms-23-11788-t001:** Tofacitinib-main case reports and clinical trials.

Disease	Drug	Study ID/Ongoing Trials	Trial Phase Study Design	Number of Patients	Study Duration	Disease Activity Score	Dosage of JAK Inhibitor/day	Concomitant Immuno-Suppressive Agents
SLE	TOFACITINIB	Hasni et al., 2021 (NCT02535689) [35,36]	Phase Ib Double blind Controlled	30	8 weeks + followed 4 weeks	SLEDAI 2K BILAG 2004 DAS28-ESR, PGA, SF-36, CLASI	5 mg × 2	NO
SLE/CLE	TOFACITINIB	NCT03288324 [49]	Phase Ib/II Open label	20	76 weeks	CLASI SLEDAI BILAG	5 mg × 2	YES
SLE/DLE	TOFACITINIB	NCT03159936 [50]	Phase I/II	5	6 months	CLASI	5 mg × 2	YES
SLE	TOFACITINIB	You et al., 2019 [45]	Case series	10	4 weeks to 12 months	SLEDAI	5 mg × 2	YES
CLE	TOFACITINIB	Bonnardeaux et al., 2022 [46]	Case series	3	1 to 7 months	CLASI	5 mg × 2	YES

Study ID = Study identifier; DLE = Discoid lupus erythematosus; CLE = cutaneous lupus erythematosus; SLEDAI 2K = Systemic Lupus Erythematosus Disease Activity Index 2000; BILAG = British Isles Lupus Assessment Group Disease Activity Index; PGA = Physician Global Assessment; DAS 28-ESR = Disease Activity Score of the 28 joints with erythrocyte sedimentation rate; SF 36 = Short Form Health Survey; CLASI = Cutaneous Lupus Erythematosus Disease Area and Severity Index.

**Table 2 ijms-23-11788-t002:** Tofacitinib-main results of clinical trials in SLE patients.

Disease Drug Study ID/Ongoing Trials	Primary Endpoints	Secondary Endpoints	Results
SLE TOFACITINIB Hasni et al., 2021 (NCT02535689) [35,36]	- safety and tolerability of tofacitinib; - rates of side effects and disease flares.	- clinical response evaluated by the impact on quality of life; - other exploratory mechanistic studies to show the effect of tofacitinib on immune dysregulation and cardiometabolic parameters necessary for the development of premature CVD.	- the study reached its primary EP regarding tofacitinib’s safety profile; - tofacitinib improved cardiometabolic and immunologic profile linked to the premature atherosclerosis in SLE by significantly decreasing STAT phosphorylation in T cells, IFNs levels in immune circulating cells (expressed by Interferon-Stimulated Genes), the percentage of LDGs and circulating NET complexes and increasing HDL-C.
SLE/CLE TOFACITINIB (NCT03288324) [49]	- oral clearance (CL/F) of the drug from plasma after oral intake.	- CLASI response; - safety profile; - modifications in SLEDAI; - BILAG scores; - SKINDEX; - global assessment score.	Estimated study completion date: June 2024.
DLE TOFACITINIB (NCT03159936) [50]	- CLASI score and safety profile of tofacitinib in DLE +/-SLE.	-	Stopped due to an insufficient number of subjects.
SLE TOFACITINIB You et al., 2019 [45]	- the level of anti-dsDNA; - C3 level; - SLEDAI-2 K; - PGA; - AEs.	-	- quickly and efficient amelioration of arthritis, but a partially improvement of skin rash; - significantly decrease of SLEDAI-2K and PGA score, but no notable serological change; anti-dsDNA levels probably based on the varied activity of SLE.
CLE TOFACITINIB Bonnardeaux et al., 2022 [46]	- efficacy of tofacitinib based on CLASI score;	- side events; - the need for adjuvant medication.	- important improvement of CLASI score.

Study ID = Study identifier; SLEDAI-2K = Systemic Lupus Erythematosus Disease Activity Index 2000; CVD = cardiovascular diseases; EP = end point; IFN = interferon; LDGs = low-density granulocytes; NETs = neutrophil extracellular traps; HDL-C = high-density lipoprotein-cholesterol; CLASI = Cutaneous Lupus Erythematosus Disease Area and Severity Index; CL/F = Apparent total clearance of the drug from plasma after oral administration; BILAG = British Isles Lupus Assessment Group; DLE = Discoid lupus erythematosus; CLE = Cutaneous lupus erythematosus; PGA = physician’s global assessment; AEs = adverse events; C3 = complement 3; dsDNA = anti-double-stranded deoxyribonucleic acid.

**Table 3 ijms-23-11788-t003:** Baricitinib-main case reports and clinical trials.

Disease	Drug	Study ID/Ongoing Trials	Trial Phase Study Design	Number of Patients	Study Duration	Disease Activity Score	Dosage of JAK Inhibitor	Concomitant Immuno-Suppressive Agents
SLE/CLE	BARICITINIB	Wallace et al., 2018 [3] (NCT02708095) Study JAHH [54]	Phase 2 trial Double blind Controlled	314	24 weeks	SLEDAI-2K	BARICITINIB 2 mg/day, or BARICITINIB 4 mg/day	YES
SLE/CLE	BARICITINIB	BRAVE I (NCT03616912) Study JAHZ [67]	Phase III Double blind Controlled	750	52 weeks	CLASI	BARICITINIB 2 mg/day, or BARICITINIB 4 mg/day	YES
SLE/CLE	BARICITINIB	BRAVE II (NCT03616964) Study JAIA [68]	Phase III Double blind Controlled	777	52 weeks	LLDAS CLASI	BARICITINIB 2 mg/day, or BARICITINIB 4 mg/day	YES
SLE	BARICITINIB	SLE-BRAVE X (NCT03843125) [69]	Phase III Double blind	1100	156 weeks	LLDAS CLASI	BARICITINIB 2 mg/day, or BARICITINIB 4 mg/day	YES
CLE	BARICITINIB	Zimmermann et al., 2019 [64]	Case series	3	3 months	R-CLASI VAS	BARICITINIB 4 mg/day	NO

Study ID = Study identifier; CLE = cutaneous lupus erythematosus; LLDAS = Lupus Low Disease Activity State; R-CLASI = revised cutaneous lupus area and severity index; VAS = visual analog scale; SLEDAI 2K = Systemic Lupus Erythematosus Disease Activity Index 2000; CLASI = Cutaneous Lupus Erythematosus Disease Area and Severity Index.

**Table 4 ijms-23-11788-t004:** Baricitinib-main results of clinical trials in SLE patient.

Disease Drug Study ID/Ongoing Trials	Primary Endpoints	Secondary Endpoints	Results
SLE/CLE BARICITINIB Wallace et al., 2018 [3] (NCT02708095) [54]	- absence of arthritis or rash at week 24 defined by SLEDAI-2K;	- number of patients with SRI-4 response after 24 weeks; - PGA; - SLEDAI-2k score; - LLDAS; - CLASI.	- significantly more patients achieved SLEDAI-2K remission of either arthritis or rash at week 24 with a high dose of baricitinib (but not baricitinib 2 mg) compared to PBO - mucocutaneous activity seen in 84% patients, but low CLASI score.
SLE/CLE BARICITINIB BRAVE I (NCT03616912) [67]	- percentage of participants with SRI-4 response for ahigh dose of baricitinib;	- percentage of SRI-4 response for low dose; - LLDAS; - FACIT-Fatigue total score, - CLASI; - tender + swollen joint count.	Study completion: March 2022 -study was stopped due to the insufficient data to support a positive benefit risk ratio.
SLE/CLE BARICITINIB BRAVE II (NCT03616964) [68]	- SRI-4 response at high dose of baricitinib;	- SRI-4 response at low dose; - LLDAS; - CLASI; - FACIT-Fatigue total score; - tender+swollen joint count.	Completed -no results yet published.
SLE BARICITINIB SLE-BRAVE X (NCT03843125) [69]	- percentage of subjects with TEAEs/AESIs/SAEs; - percentage of patients with temporary/permanent discontinuations of baricitinib;	- SRI-4 response; - LLDAS; - CLASI total score. - SELENA-SLEDAI flare index flare rate; - tender + swollen joint count, - fluctuations in SLICC/ACRdamage index total score.	Study completion: March 2022 -study terminated as there was no enough data to support a positive benefit risk ratio.
CLE BARICITINIB Zimmermann et al., 2019 [64]	- R-CLASI response with improvement of cutaneous lupus lesions; - reduction of pain due to skin and joint implication assessed by VAS; - variation of type I IFN signature in blood;-fibroblasts response to cold exposure.	-	- notable improved cutaneousmodifications as measured by R-CLASI after 3 months; - pain accompanying arthritis and skin lesions not completely remitted, in contrast to the results on cutaneous signs (one patient with complete relief of skin and joint pain, whereas in 2 patients, pain associated with joint inflammation was partially diminished as measured by VAS); - inhibition of systemic type I IFN activation in blood; - cold generated a stress response in patient’s fibroblasts; - reduction of disease flares.

Study ID = Study identifier; CLE = cutaneous lupus erythematosus; PBO = placebo; SRI-4 = SLE Responder Index 4; IFN = interferon; LLDAS = Lupus Low Disease Activity State; SELENA = Annualized Safety of Estrogens in Lupus Erythematosus National Assessment; CLASI = Cutaneous Lupus Erythematosus Disease Area and Severity Index; FACIT-Fatigue = Functional Assessment of Chronic Illness Therapy-Fatigue; SLEDAI-2K = Systemic Lupus Erythematosus Disease Activity Index 2000; TEAEs = Treatment-Emergent Adverse Events; AESIs = Adverse Events of Special Interest; SAEs = Serious Adverse Events; R-CLASI = revised cutaneous lupus area and severity index; VAS = visual analog scale; PGA = physician’s global assessment.

**Table 5 ijms-23-11788-t005:** New JAK inhibitors for SLE patients and the main results of clinical trials.

Disease Drug Study ID/Ongoing trials	Number of Patients Study Duration Disease Activity Score	Dosage of JAK Inhibitor	Results
SLE SOLCITINIB Kahl et al., 2019 [75] (NCT01777256) [77]	- 51; - 16 weeks; - SELENA-SLEDAI.	Solcitinib 50/100/200/400 mg × 2/day vs. PBO	- no significant outcome on mean interferon transcriptional biomarker expression (all panels); - discontinued because of safety issues (severe drug reaction).
SLE BREPOCITINIB (NCT03845517) [80]	- 448; - 56 weeks; - LLDAS, - CLASI.	Brepocitinib 15 mg/30 mg/45 mg vs. PBO	Estimated study completion date: August 2023; -no results yet published.
SLE UPADACITINIB (NCT03978520) SLEek [83]	- 341; - 48 weeks; - SELENA-SLEDAI, LLDAS.	Upadacitinib dose A or dose B vs. PBO vs. Elsubrutinib	Estimated completion date: July 2022; -no results yet published.
SLE/DLE R333 (NCT01597050) [79]	- 54; - 4 weeks; - Total Combined Erythema and Scaling Score.	R333 6% (60 mg/g) × 2/day	Presto et al. concluded that R333 treatment did not show a significant improvement in dermal modifications [78].
CLE FILGOTINIB (NCT03134222) [81]	- 47; - 24 weeks; - CLASI.	Filgotinib 200 mg/day vs. Lanraplenib 30 mg /day PBO	Completed -primary endpoint (CLASI activity score) not met [87].
LMN FILGOTINIB (NCT03285711) [82]	- 9; - 32 weeks.	Filgotinib 200 mg/day vs. Lanraplenib 30 mg/day	Completed -limited results due to the small number of patients [88].
SLE DEUCRAVACITINIB (NCT03252587) [85]	- 363; - 56 weeks.	Deucravacitinib dose 1 or dose 2 or dose 3 vs PBO	Completed -no results yet published.
SLE DEUCRAVACITINIB (NCT03920267) [86]	- 261; - 178 weeks; - LLDAS, CLASI, BICLA	Deucravacitinib dose 1 or dose 2 or dose 3	Estimated study completion date: November 2023.

Study ID = Study identifier; LLDAS = Lupus Low Disease Activity State; CLASI = Cutaneous Lupus Erythematosus Disease Area and Severity Index; SELENA = Annualized Safety of Estrogens in Lupus Erythematosus National Assessment; SLEDAI = Systemic Lupus Erythematosus Disease Activity Index; BICLA = Based Combined Lupus Assessment; LMN = lupus membranous nephropathy; CLE = cutaneous lupus erythematosus; DLE = Discoid lupus erythematosus; PBO = placebo.

## Data Availability

Not applicable.

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
