# Peer review of "Why Do We Need JAK Inhibitors in Systemic Lupus Erythematosus?"

_ijms, 2022, doi:10.3390/ijms231911788_

Round 1

Reviewer 1 Report

The authors review published and ongoing studies regarding the role of JAK inhibitors in systemic lupus erythematosus. The review is comprehensive. I have severeal suggestions for the authors.

1. Line 161. The full form of abbreviation CLASI is Cutaneous Lupus Erythematosus Disease Area and Severity Index. The full form should be described before the abbreviation.

2. Table 2, 4, and 6 look disorganized. It is unnecessary to list details such as all inclusion and exlusion criteria in all clinical trials. Instead, the authors may summary main findings in each clinical trial. Also, it is unnecessary to create abbrevations for these JAK inhibitors (ex. TOFA for tofacitinib)

3. Line 295. The full form of DRESS (drug reaction with eosinophilia and systemic symptoms) should be mentioned before the abbreviation.

4. Both table 5 and table 6 describe clinical trials of new JAK inhibitors for SLE (The titles of the two tables are almost the same). The authors may consider summarizing main findings of these clinical trial using one table.

5. Line 352. There is a typo "changeling". 

Author Response

Dear Reviewer,

Thank you very much for the attention given to this work and for the requested changes that surely improved the quality of the article. Following the instructions received, I made the following changes, as follows

- For the acronyms CLASS and DRESS, we have added the full name before the abbreviation.

- We kept the full names to all JAK inhibitors without abbreviating them.

- In Table 2 and Table 4 we have summarized the information; in addition we have removed the inclusion and exclusion criteria.

- Also, in all tables we have updated the list of abbreviations located below the table. Therefore, the typo in line 352 has been removed.

- We have merged Table 5 with Table 6. For this we had to delete from the text: trial phase, study design, inclusion criteria, exclusion criteria, primary and secondary endpoints. We kept important elements such as the Results. 

- We also changed the title of Table 5.

- Obviously, we have removed from the text the references of Table 6.

- Going through the literature again, we have added 2 more references, of course respecting the order of appearance in the text.

We hope that these changes satisfy your requirements and we look forward to other suggestions if you consider them appropriate.

Sincerely,

The authors

Reviewer 2 Report

The review is well written and organised. It provides systematic information on the use of JAK inhibitors in lupus. However, I highly recommend adding a figure regarding the JAK inhibitors  mechanism of action.

Author Response

Dear Reviewer,

Thank you very much for the attention given to this work and for the requested changes that surely improved the quality of the article. Following the instructions received, we introduced figure 1 which shows the mechanism of action of JAK inhibitors as well as the main molecules in clinical trials

We hope that these changes satisfy your requirements and we look forward to other suggestions if you consider them appropriate.

Sincerely,

The authors

Reviewer 3 Report

I considered the manuscript entitled “Why do we need jak inhibitors in systemic lupus erythematosus?” by Patricia Richter, et al, that is intended to be published in IJMS journal.

The review deals with a field of highly interest that is the discovery or implementation to the clinics of new drugs for SLE. JAK/STAT inhibitors are being studied for this disease from time ago but few sound has been heard in medical literature apart from the specific publications. This review is just informative, but it gives almost all the data that is currently known. The narrative is balanced in this kind of reviews.

To me, a Figure where the intracellular mechanisms are described and pictured must be introduced. The review is addressed to Clinicians but a glance to a Figure is also highly informative.

Author Response

(The authors gave the same response as above.)

Round 2

Reviewer 1 Report

The authors and responded to my previous comments/suggestions and revised the manuscript accordingly. I think the manuscript is acceptable in its current form.

Author Response

Dear Reviewer,

Thank you very much for the suggestions, for the time spent on this work and for agreeing to continue the publication process.

Best regards,
Authors

Reviewer 2 Report

No other concerns. 

Author Response

(The authors gave the same response as above.)

Reviewer 3 Report

No further

Author Response

(The authors gave the same response as above.)
